# Facing the Realities of Pragmatic Design Choices in Environmental Health Studies: Experiences from the Household Air Pollution Intervention Network Trial

**DOI:** 10.3390/ijerph19073790

**Published:** 2022-03-23

**Authors:** William Checkley, Shakir Hossen, Ghislaine Rosa, Lisa M. Thompson, John P. McCracken, Anaite Diaz-Artiga, Kalpana Balakrishnan, Suzanne M. Simkovich, Lindsay J. Underhill, Laura Nicolaou, Stella M. Hartinger, Victor G. Davila-Roman, Miles A. Kirby, Thomas F. Clasen, Joshua Rosenthal, Jennifer L. Peel

**Affiliations:** 1Division of Pulmonary and Critical Care, School of Medicine, Johns Hopkins University, Baltimore, MD 21287, USA; shossen1@jhu.edu (S.H.); lindsayu@wustl.edu (L.J.U.); l.nicolaou@jhu.edu (L.N.); 2Center for Global Non-Communicable Disease Research and Training, School of Medicine, Johns Hopkins University, Baltimore, MD 21287, USA; 3Faculty of Infectious and Tropical Diseases, London School of Tropical Medicine and Hygiene, London WC1E 7HT, UK; ghislaine.rosa@lshtm.ac.uk; 4Nell Hodgson Woodruff School of Nursing, Emory University, Atlanta, GA 30332, USA; lisa.thompson@emory.edu; 5Epidemiology and Biostatistics Department, University of Georgia, Athens, GA 30606, USA; john.mccracken@uga.edu; 6Center for Health Studies, Universidad del Valle de Guatemala, Guatemala City 01015, Guatemala; adiaz@ces.uvg.edu.gt; 7ICMR Center for Advanced Research on Air Quality, Climate and Health, Department of Environmental Health Engineering, Sri Ramachandra Institute for Higher Education and Research, Chennai 600116, India; kalpanasrmc@ehe.org.in; 8Division of Healthcare Delivery Research, Medstar Health Research Institute, Hyattsville, MD 20782, USA; suzanne.m.simkovich@medstar.net; 9Division of Pulmonary and Critical Care Medicine, Georgetown University, Washington, DC 20007, USA; 10Latin American Center of Excellence on Climate Change and Health, Universidad Peruana Cayetano Heredia, Lima 15102, Peru; stella.hartinger.p@upch.pe; 11Cardiovascular Imaging and Clinical Research Core Laboratory, Cardiovascular Division, Department of Medicine, Washington University in St. Louis, St. Louis, MO 63110, USA; vdavila@wustl.edu; 12Department of Global Health and Population, Harvard T.H. Chan School of Public Health, Boston, MA 02115, USA; mkirby@hsph.harvard.edu; 13Gangarosa Department of Environmental Health, Rollins School of Public Health, Emory University, Atlanta, GA 30322, USA; thomas.f.clasen@emory.edu; 14Fogarty International Center, National Institutes of Health, Bethesda, MD 20892, USA; rosenthj@ficod.fic.nih.gov; 15Department of Environmental and Radiological Health Sciences, Colorado State University, Fort Collins, CO 80523, USA; jennifer.peel@colostate.edu

**Keywords:** household air pollution, randomized trials, efficacy, effectiveness

## Abstract

**Objective:** Household Air Pollution Intervention Network (HAPIN) investigators tested a complex, non-pharmacological intervention in four low- and middle-income countries as a strategy to mitigate household air pollution and improve health outcomes across the lifespan. Intervention households received a liquefied petroleum gas (LPG) stove, continuous fuel delivery and regular behavioral reinforcements for 18 months, whereas controls were asked to continue with usual cooking practices. While HAPIN was designed as an explanatory trial to test the efficacy of the intervention on four primary outcomes, it introduced several pragmatic aspects in its design and conduct that resemble real-life conditions. We surveyed HAPIN investigators and asked them to rank what aspects of the design and conduct they considered were more pragmatic than explanatory. **Methods:** We used the revised Pragmatic Explanatory Continuum Indicator Summary (PRECIS-2) to survey investigators on the degree of pragmatism in nine domains of trial design and conduct using a five-point Likert rank scale from very explanatory (1) to very pragmatic (5). We invited 103 investigators. Participants were given educational material on PRECIS-2, including presentations, papers and examples that described the use and implementation of PRECIS-2. **Results:** Thirty-five investigators (mean age 42 years, 51% female) participated in the survey. Overall, only 17% ranked all domains as very explanatory, with an average (±SD) rank of 3.2 ± 1.4 across domains. Fewer than 20% of investigators ranked eligibility, recruitment or setting as very explanatory. In contrast, ≥50% of investigators ranked the trial organization, delivery and adherence of the intervention and follow-up as very/rather explanatory whereas ≤17% ranked them as rather/very pragmatic. Finally, <25% of investigators ranked the relevance of outcomes to participants and analysis as very/rather explanatory whereas ≥50% ranked then as rather/very pragmatic. In-country partners were more likely to rank domains as pragmatic when compared to investigators working in central coordination (average rank 3.2 vs. 2.8, respectively; Wilcoxon rank-sum *p* < 0.001). **Conclusion:** HAPIN investigators did not consider their efficacy trial to be rather/very explanatory and reported that some aspects of the design and conduct were executed under real-world conditions; however, they also did not consider the trial to be overly pragmatic. Our analysis underscores the importance of using standardized tools such as PRECIS-2 to guide early discussions among investigators in the design of environmental health trials attempting to measure efficacy.

## 1. Introduction

Randomized controlled trials (RCTs) stand atop the pinnacle of study designs used by public health scientists to generate causal evidence about whether interventions work to improve health outcomes [1]. The appeal of RCTs lies in their experimental nature and simplicity, whereby all potential confounders—both measured and unmeasured—are equally balanced between trial arms through the process of randomization. Even the most carefully conducted observational studies, in contrast, cannot rule out residual confounding due to inadequately measured or unmeasured factors. This issue is particularly important for environmental health trials, where it is often difficult to disentangle the effects of environmental exposures from other influencing factors such as poverty [2]. However, the use of RCTs in environmental health remains relatively uncommon. Indeed, one scoping review found that 0.6% of environmental health publications between 2000 and 2015 were RCTs of an intervention to reduce exposure and improve health [3].

There are several factors in the design, conduct and analysis of RCTs that may affect the interpretation of findings. One such choice is the analytical strategy used to measure an intervention effect. Historically, environmental health scientists have relied on exposure-response (E-R) analysis for inference, even if data is obtained from an RCT. However, most environmental health RCTs should be analyzed using an intention-to-treat (ITT) approach in which all participant outcomes are included and analyzed according to their original randomization assignment, regardless of intervention fidelity and adherence; they may also be analyzed per-protocol, whereby the treatment effect is analyzed based on adherence with the intervention. In such cases, however, the risk of confounding is present due to breaking randomization. Environmental health trials also commonly include E-R analyses that can help reduce misclassification of exposure due to inadequate adherence with the intervention, again at the cost of breaking randomization and the attendant risk of unmeasured confounding [4]. Thus, ITT analyses remain the most conservative approach in environmental health RCTs to make recommendations for policy or practice decisions [5,6].

The interpretation of an RCT can also be affected by whether it is designed and conducted primarily as an explanatory trial to measure intervention efficacy under controlled conditions or a pragmatic trial to assess intervention effectiveness in real-world conditions [7,8]. Under controlled conditions, such as carefully identifying optimal settings and study populations, setting narrow eligibility criteria, designing the intervention, taking charge of its delivery and adhering to established protocols to ensure uptake, investigators can answer whether an intervention strategy is efficacious under those precise conditions if statistically superior to the control strategy. Such trials with an explanatory focus are useful in determining proof of principle, where the investigators want to reduce the risk of complications, such as a poorly designed intervention, poor delivery, or adherence, i.e., factors that often render it impossible to answer questions about intervention effects. On the other hand, RCTs that incorporate real-world conditions into their design and conduct are more pragmatic in that they allow for estimates of an intervention effect under programmatic conditions which are less controlled but often more generalizable. A positive effect from highly pragmatic RCTs suggests that interventions are effective under real-world conditions. However, a null finding in a highly pragmatic RCT does not necessarily mean that an intervention was not efficacious [7]. Understanding how pragmatic choices affect the design and conduct of an efficacy RCT is therefore paramount.

In this paper, we sought to evaluate perceptions among investigators regarding the degree of pragmatism in the Household Air Pollution Intervention Network (HAPIN) trial. We used the revised Pragmatic Explanatory Continuum Indicator Summary (PRECIS-2) to survey HAPIN investigators on the degree of pragmatism in nine domains of trial design and conduct [9]. PRECIS-2 is a tool to help trialists make design decisions consistent with the intended purpose of their trial. PRECIS-2 evaluates nine domains of trial design and conduct, namely eligibility, recruitment, setting, organization, flexibility in delivery, flexibility in adherence, follow-up, primary outcome and analysis, and uses a five-point Likert rank scale from very explanatory (1) to very pragmatic (5) to rank each of these domains. Therefore, efficacy trials would be expected to rank at or near 1 whereas effectiveness trials would be expected to rank at or near 5. While the PRECIS-2 tool was designed to assist investigators at the design stage of a trial, it can also be used retrospectively, as used here.

## 2. Materials and Methods

### 2.1. Study Setting

We conducted a multi-country RCT to evaluate the effect of a household air pollution (HAP) intervention on four primary health outcomes in 3200 households in 10 resource-poor settings of four low- and middle-income countries: one district in Jalapa, Guatemala; two districts in Tamil Nadu, India; one district in Kayonza, Rwanda; and six provinces in Puno, Peru. The trial is described in detail elsewhere [10]. Briefly, households in the intervention group were randomly assigned to receive a liquefied petroleum gas (LPG) stove, continuous fuel delivery and behavioral reinforcements for 18 months, whereas those in the control group were asked to continue with usual cooking practices. Control homes received periodic compensation to offset the economic benefit of the intervention. Each intervention research center (IRC) screened for pregnant women aged 18–34 years who were at 9–19 weeks gestation. In approximately 15% of households, we also enrolled an older adult woman aged 40–79 years. Households were individually randomized across the 10 strata. Pregnant women were followed until birth and their offspring were then followed for 12 months. Older adult women were followed for 18 months.

The four primary outcomes for this trial were birth weight, severe pneumonia and stunting in children during the first year of life, and blood pressure in older adult women [10]. We conducted three home or clinic visits to evaluate personal HAP exposures and fetal biometrics in pregnant women, and three home visits to measure personal HAP exposures and blood pressure in older adult women (at enrollment and two visits after randomization); hospital visits to weigh newborns within 24 h of birth or for active surveillance of severe pneumonia in children; quarterly home visits for anthropometry and indirect assessment of HAP exposure in children [11]; and personal HAP exposures [11] and blood pressure [10] in older adult women. We used stove use monitors in all biomass-burning stoves of intervention households to monitor adherence to the LPG stove [12]. Additional home visits were conducted to deliver LPG fuel tanks to intervention homes and provide behavioral reinforcement based on stove use monitor data that indicated that a biomass stove was used [13].

### 2.2. Study Design

We invited 103 HAPIN investigators to fill out the PRECIS-2 survey. This list included all investigators who were identified as authors by the Principal Investigators or site investigators and did not include field or laboratory staff at the IRCs. We recruited participants through email invitations and during weekly steering committee meetings midway through the trial. Those who agreed to participate were given instructions and educational material in English on how to complete the PRECIS-2 survey: two papers that described the use and implementation of PRECIS-2 [9], a presentation that summarized how to rank trials using PRECIS-2 (Online Supplement), and four trials that served as an example to help standardize ranking across the nine domains [14,15,16,17]. One investigator (WC) conducted two one-hour training sessions on the PRECIS-2 tool and how to rank the domains with all investigators during recruitment. However, investigators who agreed to participate in this survey were ultimately responsible for reading through the papers, evaluating the examples, and asking questions about the tool if there were any concerns. This survey of participating investigators was judged to be exempt from human subjects’ regulations.

Participating investigators entered their answers to the survey in an online database (Qualtrics, Provo, UT, USA). We collected basic demographic information—including the investigator’s name, age, role on the project including whether they worked primarily in one of the central coordination cores or at one of the IRCs—and answers to the PRECIS-2 survey. We did not ask participants about their qualifications or years of training. Given that there were four primary outcomes in the HAPIN trial, we asked investigators to provide four separate answers to the questions regarding the outcome and analysis domains. Investigators provided responses to this survey between 5 July 2019, and 2 May 2020, approximately 1–2 years after the start of the trial but before follow-up activities were completed.

### 2.3. Definitions

The nine domains of design and conduct in PRECIS-2 include eligibility, recruitment, setting, organization, flexibility in delivery, flexibility in adherence, follow-up, primary outcome and primary analysis. Responses to the PRECIS-2 survey follow a five-point Likert scale [9,18], ranging from very explanatory with a rank of 1 to very pragmatic with a rank of 5 (Figure 1). The nine domains of design and conduct in PRECIS-2 were selected by an expert panel of trialists after careful analyses that included two rounds of the Delphi method with authors who were not involved in the original development of PRECIS, focus groups with trialists including the original developers of PRECIS, and external validation with 19 international trialists [9].

There is no established convention on how to summarize the ranks in PRECIS-2. Here we used measures of central tendency (mean, median or both) as summary statistics. We also grouped answers on either end of the scale, “Very pragmatic” (rank = 5) and “Rather pragmatic” (rank = 4) as pragmatic and “Rather explanatory” (rank = 2) and “Very explanatory” (rank = 1) as explanatory.

### 2.4. Statistical Analysis

The primary analytical objective was to summarize investigator responses to the nine domains of the PRECIS-2 survey [9]. The primary hypothesis is that the HAPIN trial, an RCT designed to test the efficacy of a household air pollution intervention strategy, would rank as rather or very explanatory. We calculated the mean (standard deviation, SD) or median (interquartile range, IQR) ranks for each domain and across all domains. For the purposes of visualization, we averaged each of the four responses for the domains pertaining to outcomes and analysis into a single response. We used the non-parametric Wilcoxon rank-sum test to compare the distribution of values between participants who belonged to the IRCs or worked in central coordination. We conducted complete case analyses only; however, there were no missing data in any of the PRECIS-2 domains and only one investigator did not provide age. All analyses and visualizations were performed in R, version 4.04, “Lost Library Book” [19].

## 3. Results

### 3.1. Participant Characteristics

A total of 35 investigators (34%) agreed to participate and responded to the online survey. The average (±SD) age was 42.0 ± 10.7 years, and 18 participants (51%) were female. A total of 21 (60%) investigators worked primarily in one IRC and the remaining 14 (40%) worked in central coordination. Of those who belonged to an IRC, we had 4 responses from Guatemala, 3 from India, 8 from Peru and 6 from Rwanda. All three trial Principal Investigators (PIs), at least one site PI from each IRC, and at least one investigator from each of the central coordination cores responded to the survey.

### 3.2. Ranking of the Nine PRECIS-2 Domains

We plotted median and IQR ranks to all domains in Figure 2. Mean (±SD) rank and median (±IQR) ranks were 3.2 ± 1.4 and 3 ± 2 across all domains, respectively. Overall, only 88 (16.7%) of the 525 responses were ranked as very explanatory, 170 (32.4%) were ranked as very/rather explanatory, and 229 (43.6%) were ranked as rather/very pragmatic.

Few investigators ranked eligibility (5 or 14.3%), recruitment (5 or 14.3%) or setting (4 or 11.4%) as very explanatory and similar proportions of investigators ranked eligibility (34.3% vs. 28.6%) and recruitment (37.1% vs. 31.4%) as either rather/very explanatory or rather/very pragmatic, respectively. A slightly larger proportion of investigators ranked setting as rather/very pragmatic (43%) than rather/very exploratory (26%). In contrast, ≥50% of investigators ranked the trial organization (19 or 54.3%), delivery (23 or 65.7%) and adherence (11 or 60%) of the intervention and follow-up (22 or 62.8%) as rather/very explanatory, whereas ≤17% ranked trial organization (5 or 14.3%), delivery (5 or 14.3%) and adherence (6 or 17.1%) to the intervention, and follow-up (4 or 14.3%) as rather/very pragmatic. Finally, <25% of investigators ranked the relevance of the four outcomes to participants (17/140 responses or 12.1%) or analysis (34/140 or 24.2%) as rather/very explanatory, whereas ≥50% ranked outcomes (93/140 or 66.4%) and analysis (79/140 or 56%) as rather/very pragmatic.

### 3.3. Ranking According to Where Investigators Worked

We plotted median and IQR ranks according to whether investigators worked primary at an IRC or not in Figure 3. On average, investigators who worked primarily in an IRC ranked the trial as more pragmatic than did those working in central coordination. Overall, the mean (± SD) ranks for all domains were 2.8 ± 1.4 vs. 3.4 ± 1.4 for investigators working in central coordination vs. those from an IRC (*p* < 0.001; Wilcoxon rank-sum test). When evaluated by domains, however, we did not find differences in mean ranks for eligibility (2.8 vs. 2.9; *p* = 0.70), recruitment (2.8 vs. 3.0; *p* = 0.74), screening (3.1 vs. 3.5; *p* = 0.33) or trial organization (2.3 vs. 2.5; *p* = 0.56) between investigators working in central coordination vs. those from IRCs, respectively. While the mean values of ranks for flexibility in adherence to the intervention (1.9 vs. 2.5; *p* = 0.20), outcomes (3.3 vs. 3.8; *p* = 0.22) or analysis (3.6 vs. 3.9; *p* = 0.29) were lower for investigators working in central coordination when compared to that of investigators from IRCs, respectively, they did not achieve statistical significance. Finally, mean values of ranks for flexibility in the delivery of the intervention (1.5 vs. 2.5; *p* = 0.01) and follow-up (1.6 vs. 2.5; *p* = 0.04) were lower for investigators working in central coordination when compared to those from IRCs.

## 4. Discussion

We surveyed HAPIN investigators regarding their perceptions of the degree of pragmatism introduced into the design and conduct of their efficacy trial and found that most did not consider it to be entirelyexplanatory when using the PRECIS-2 scale. Indeed, the average rank for the nine domains of design and conduct was 3, in contrast to a value of 1 or 2 expected for explanatory trials. Conditions that HAPIN investigators considered less than very explanatory included the study setting, participant selection and recruitment. However, investigators did not consider the trial to be overly pragmatic either, suggesting an attempt to balance design choices between ideal and real-life conditions.

It is important to acknowledge that the PRECIS-2 tool was designed to help trialists frame the conversation around study design and conduct when the primary focus is to develop highly pragmatic trials under real-world conditions [9]. Indeed, the PRECIS-2 tool does not appear to have been previously used to aid the design and conduct of, or rate, efficacy trials. However, the same principle applies: it is important to frame the conversation among investigators around the amount of pragmatism introduced into an efficacy trial to avoid potential pitfalls in study design and conduct. In efficacy trials, early discussions using a structured approach such as that outlined in PRECIS-2 can help investigators choose the best combination of controlled conditions to test an intervention.

HAPIN was designed to be a proof-of-concept, efficacy trial. Indeed, the investigators conducted at least 18 months of formative field research [20,21,22,23,24] or leveraged ongoing research [25] to identify study settings with low population density and low ambient air pollution. Formative activities included training personnel in ultrasound measurements to assess gestational period that met eligibility criteria [26], developing procedures for stove installation and training in stove use [10], assessing methods of fuel delivery to assure a continuous fuel supply, developing and implementing behavioral messages for intervention adherence, including monitoring temperature of biomass fuel stoves in intervention homes [13] and collecting outcome data within specific time windows [10]. All these activities are in line with the conduct of an efficacy trial.

However, some aspects of the HAPIN trial design and conduct, as noted above, were reported as not meeting ideal or controlled conditions by the investigators. This is potentially important because the introduction of too much pragmatism in an efficacy trial may ultimately affect its interpretation. Indeed, the treatment effect is an extrinsic property that depends not only on the intervention being tested but also on participant selection and on the circumstances in which the intervention is being measured [7]. For example, a clean energy intervention is likely to be most successful at improving health outcomes if participant exposures to air pollution are reduced to levels below recommended air quality targets, e.g., the World Health Organization interim target for mean annual concentrations of PM_2.5_ < 35 μg/m^3^ [27]. However, if ambient air pollution at the study site exceeds recommended air quality targets, then how can the efficacy of the HAP intervention be tested? Intentionally, the HAPIN trial settings were selected in areas with low ambient air pollution. Furthermore, the policy context for the intervention was a motivating factor for the conduct of this trial and a driving force in local and international discussions. Indeed, questions such as “When and what can we present to local policy makers?” and “Will this help motivate subsidies for fuel at the national level?” were a frequent topic of conversation among HAPIN investigators. While HAPIN investigators aimed for equipoise before and throughout the rollout of this trial, many investigators believed that the intervention could make a difference in health outcomes if reductions in personal exposures to HAP were reduced sufficiently and sustained over an extended period of time. Ongoing conversations in the early stages of the trial likely reflect a pragmatic, foundational set of expectations and may have played a role in the investigators’ responses to the PRECIS-2 survey. Given the paucity of environmental health RCTs, environmental health scientists have historically relied on exposure–response analyses of observational data rather than on intention-to-treat analyses of RCTs. This may explain why many investigators of the HAPIN trial in associated conversations put equal weight into the findings of an exposure–response analysis as they would to the one based on intention-to-treat.

Another condition that can affect the measurement of a treatment effect is the choice of usual care for the control group [28]. In HAP trials, usual care means allowing people to continue with their usual cooking practices. For example, 1% and 6% control homes in India and Peru had LPG stoves at baseline whereas none did in Rwanda and Guatemala [11]. The choice of usual care for the control group may have prompted some HAPIN investigators to consider the study setting less than very explanatory. While common in effectiveness trials, usual care in efficacy RCTs can be problematic because it does not require a uniform approach to what the control group receives in lieu of the intervention. Usual care may also vary substantially across settings [28], further complicating the interpretation of findings in multi-center studies. In HAP trials, the lack of a uniform approach can result in significant variability in cooking practices within and between control participants that may be difficult to measure, reducing exposure contrasts and therefore limiting the value of the control group as a comparator. This may be alleviated by monitoring cooking practices longitudinally in the control households using stove use monitors in all biomass-burning stoves. The problem of usual care in HAP trials is further compounded by the unavoidable lack of blinding of the intervention. Indeed, in learning that the HAP intervention is aimed at reducing exposures to improve health outcomes, control participants may consciously or unconsciously change their cooking practices to reduce their own exposures or even choose to use the same treatment that was given to intervention participants [29].

Investigators from the IRCs who worked directly in the implementation of the RCT at their sites considered it to be more pragmatic than investigators working in central coordination. Indeed, the PRECIS-2 survey also identified differences in ranking between in-country partners and investigators working in central coordination. While developing the protocol for the trial, HAPIN investigators debated to what degree they would allow for variations according to local context across the four international settings. Although trial standards were set centrally in coordination with in-country partners, there was a necessary amount of customization to the intervention when executing the protocol at the four international locations. Indeed, during weekly calls, HAPIN investigators listened to how in-country partners often had different challenges that required local solutions even when the basic protocol was shared: from stove type and cylinder size, to a pledge to use the intervention stove administered to the intervention households and behavioral reinforcement methods, to the pressures of having to enroll quickly, to which biomass stoves were monitored, to variations in clinical surveillance systems, background ambient conditions, housing densities, housing types, family age structures, access to health and other services, even types of solid fuels used, and location of cooking. Frequent needs to adjust local activities to the conditions posed by the COVID-19 pandemic toward the latter end of the trial also inevitably reinforced recognition of the real-world conditions in which this efficacy trial took place. Therefore, it is easy to understand how in-country partners working day-to-day with all the inherent messiness of the field would see their work as more pragmatic than the people who designed the trial and managed it mostly from afar.

While HAPIN investigators did not consider their trial to be very explanatory, they also did not consider it to be entirely pragmatic either. The trial organization was reported as being rather to very explanatory by most investigators. LPG fuel was delivered to participants’ homes, removing the barrier of transportation to pick up fuel and bring it home, thus making delivery of the intervention more explanatory than pragmatic. Finally, participants were followed and monitored closely, and unsurprisingly, here investigators reported follow-up and adherence to the intervention as rather or very explanatory.

This analysis is an important reminder about the use of standardized tools such as PRECIS-2 to frame the conversation about aspects of design and conduct of a trial among investigators [30], whether it is an efficacy or effectiveness trial. The PRECIS-2 survey also provides good interrater reliability and modest discriminant validity in these evaluations [31]. Moreover, a careful review of the trial components before its execution may help to make some adjustments to the study protocol. However, our study also has some potential shortcomings. First, we conducted the survey midway through the trial rather than at the time of study design but before investigators were unblinded to the results of the trial. While there may be some limitations in the use of the PRECIS-2 tool for the assessment of a trial, a recent study found this tool to be effective in a retrospective evaluation of trials and systematic reviews [32]. Although our midway analysis did not allow us to make protocol adjustments in the study design phase, it was far enough along in design and execution that investigators were implementing the efforts and had a concrete sense of the compromises made at multiple levels. Moreover, this survey was conducted before the investigators saw any results, and the views of the survey are not influenced by how the results matched with any expectations. Second, only a third of investigators agreed to respond to the survey, which may have led to a potentially biased result either from measurement or selection bias. However, previous studies on surveys have identified that response rates are not indicative of the quality of responses [33,34]. While identities of the responders were not shared beyond the lead analyst for this study, some in-country investigators may have not participated in this study for fear that it could affect their jobs if they did not answer the questions in the way that met the objectives of the trial or for fear of being accused of not properly conducting trial procedures. They may have also not chosen to participate because of a language barrier. However, it is also important to note that while HAPIN has 103 investigators listed as contributors, only approximately 50 participated actively in weekly meetings. Therefore, our actual participation rate may be an underestimate. Third, we may have not adequately trained investigators to respond to the Likert scale in a standardized manner. While PRECIS-2 was developed to aid communication between investigators, it does not provide instructions on how to best train investigators to ensure consistent scoring. We have identified this as an area for future improvement when conducting PRECIS-2 surveys. Fourth, we did not include participants’ qualifications or years of training which may affect how the PRECIS-2 survey was scored.

## 5. Conclusions

In summary, HAPIN investigators did not consider this efficacy trial to be overly explanatory and reported that some aspects of the design and conduct were executed under real-world conditions. However, investigators did not consider the trial to be overly pragmatic either. While it is unclear how an average rank of 3 in the PRECIS-2 scale may ultimately affect the interpretation of findings for an efficacy trial, our analysis demonstrates how tools such as PRECIS-2 can provide valuable context regarding the level of pragmatic design introduced into an efficacy trial. Furthermore, it underscores the importance of using these standardized tools to guide early discussions among investigators in the design of environmental health trials so that study results align accordingly with original study intentions to measure efficacy or effectiveness.

## Figures and Tables

**Figure 1 ijerph-19-03790-f001:**
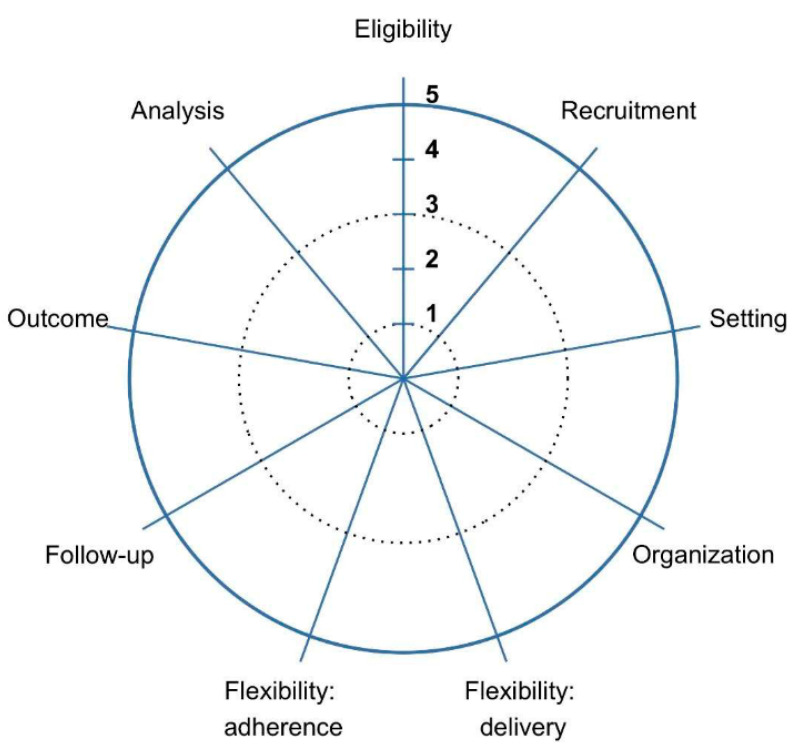
The Pragmatic Explanatory Continuum Indicator Summary 2 (PRECIS-2) wheel ranking the degree of pragmatism for the nine PRECIS-2 domains. The PRECIS-2 nine domains include: eligibility, recruitment, setting, organization, flexibility in delivery, flexibility in adherence, follow-up, primary outcome and primary analysis. It uses a five-point Likert rank scale from very explanatory (1) to very pragmatic (5) to rank each of these domains.

**Figure 2 ijerph-19-03790-f002:**
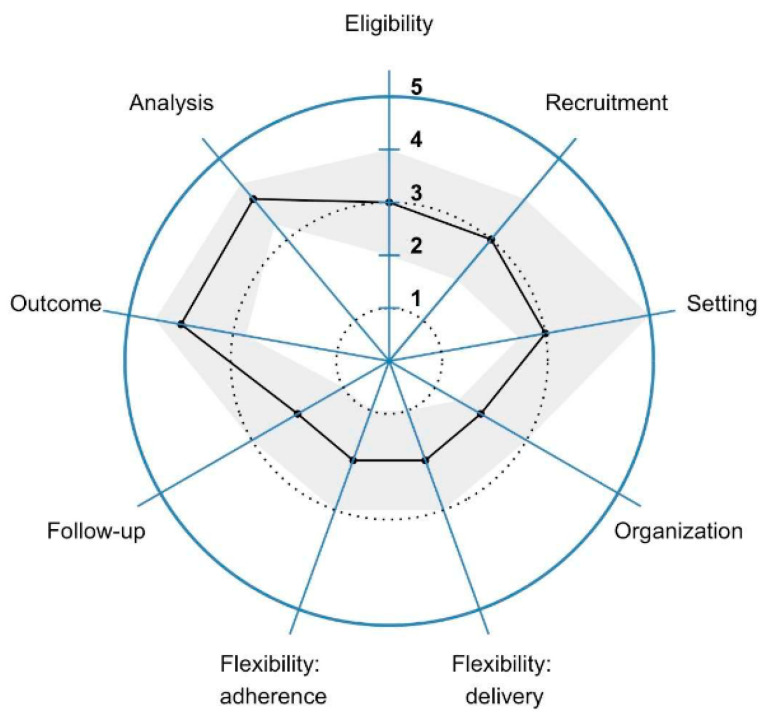
The Pragmatic Explanatory Continuum Indicator Summary 2 (PRECIS-2) wheel ranking the degree of pragmatism for the nine PRECIS-2 domains. In this figure, we report median (black line) and interquartile range (gray shaded area) ranks of the responses for of the 35 Household Air Pollution Intervention Network (HAPIN) investigators. It uses a five-point Likert rank scale from very explanatory (1) to very pragmatic (5) to rank each of these domains.

**Figure 3 ijerph-19-03790-f003:**
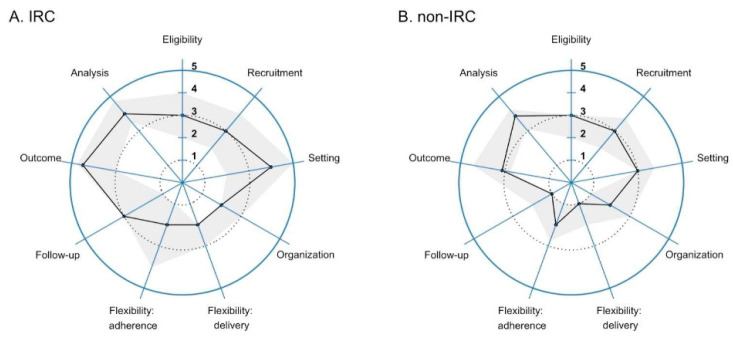
The Pragmatic Explanatory Continuum Indicator Summary 2 (PRECIS-2) wheel ranking the degree of pragmatism for the nine PRECIS-2 domains stratified by whether the investigator worked in an Intervention Research Center (IRC) or in central coordination. In this figure, we show the median and interquartile range for the responses of the 35 Household Air Pollution Intervention Network (HAPIN) investigators. Panel (**A**) plots the responses for IRC investigators (n = 21) and Panel (**B**) for investigators working in central coordination (n = 14). We report median (black line) and interquartile range (gray shaded area) ranks of the responses. It uses a five-point Likert rank scale from very explanatory (1) to very pragmatic (5) to rank each of these domains.

## Data Availability

Data are available from the corresponding author upon request.

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
