# Peer review of "Facing the Realities of Pragmatic Design Choices in Environmental Health Studies: Experiences from the Household Air Pollution Intervention Network Trial"

_ijerph, 2022, doi:10.3390/ijerph19073790_

Round 1

Reviewer 1 Report

This paper is focused on the design and conduct of a survey about the household air pollution. Sixteen main authors from varied countries are involved revealing the interest of the subject. However, major changes must be introduced prior to its publication in International Journal of Environmental Research and Public Health.

The main inconvenience is that the household air pollution survey is mixed with the management of this survey. A short standalone section about this survey and its results may be useful to place the reader in this context. The rest of the paper should be focused on the survey itself, but not on the survey content.

Minor remark.

Lines 208-209. Since the median is a robust statistic against the mean. What is the reason to select the median with more than 10 responses?

Author Response

Reviewer 1

This paper is focused on the design and conduct of a survey about the household air pollution. Sixteen main authors from varied countries are involved revealing the interest of the subject. However, major changes must be introduced prior to its publication in International Journal of Environmental Research and Public Health.

R1Q1. The main inconvenience is that the household air pollution survey is mixed with the management of this survey. A short standalone section about this survey and its results may be useful to place the reader in this context. The rest of the paper should be focused on the survey itself, but not on the survey content.

R1A1.  We apologize but the question posed by the reviewer is unclear to us as written. This manuscript describes the results of the PRECIS-2 survey, which is a survey designed to assess the degree of pragmatism in trials. We administered this survey in the context of the Household Air Pollution Intervention Network (HAPIN) trial which we summarize in detail in the methods and note the survey respondents were study investigators. The survey is not a household air pollution survey and is not assessing environmental outcomes. Instead, we describe the survey content and the context in the last paragraph of the introduction, further details of its administration in the methods, and the results of the survey in the results section (see R2A2 as we have added additional clarifying information in the Introduction section). The survey content itself is provided in the Online Supplement (see ppt).     

R1Q2. Minor remark. Lines 208-209. Since the median is a robust statistic against the mean. What is the reason to select the median with more than 10 responses?

R1A2. We thank the reviewer for this question. We agree with this reviewer that this sentence as written adds confusion and have removed it from the methods because we present both means and medians in this manuscript.  All measures of central tendency (whether mean or median) were calculated for responses with more than 10 values.

In the methods we now say:

“There is no established convention on how to summarize the ranks in PRECIS-2. Here we used measures of central tendency (mean, median or both) as summary statistics. We also grouped answers on either end of the scale, “Very pragmatic” (rank=5) and “Rather pragmatic” (rank=4) as pragmatic and “Rather explanatory” (rank=2) and “Very explanatory” (rank=1) as explanatory.

Statistical analysis

The primary analytical objective was to summarize investigator responses to the nine domains of the PRECIS-2 survey (Loudon et al. 2015). The primary hypothesis is that the HAPIN trial, an RCT designed to test the efficacy of a household air pollution intervention strategy, would rank as rather or very explanatory.  We calculated the mean (standard deviation, SD) or median (interquartile range, IQR) ranks for each domain and across all domains.”

Reviewer 2 Report

It’s a very interesting topic to study that use Pragmatic Explanatory Continuum Indicator Summary on investigators to evaluate the pragmatic and explanatory of the investigation trial. I believe it’s the first study within my sight. I suggest to publish it with minor revisions.

Major concern:

The authors have admitted that the PRECIS-2 tool does not appear to have been previously used to aid the design and conduct of, or rate, efficacy trials. It would be more convincible that author can either list some evidences or indicate some similar studies or close studies to demonstrate that the use of PRECIS-2 tools herein is proper.

Minor revision:

Some readers may not very familiar with PRECIS-2. It’s better to explain more in the introduction to help readers better understand.

Line 136: How many control homes have LPG stove by themselves. That may important to know how their usual cooking practices are.

Line 147: How is personal HAP and indirect assessment of HAP exposure in children measured?

Line 184: How are the nine domains of design in PRECIS-2 chosen?

Line 248

Authors compared ranking according to where investigators worked. Did author consider other factors such as age, sex, country, location? It would be better to show the interaction between these factors and comparison of rankings across these factors

Author Response

Reviewer 2

It’s a very interesting topic to study that use Pragmatic Explanatory Continuum Indicator Summary on investigators to evaluate the pragmatic and explanatory of the investigation trial. I believe it’s the first study within my sight. I suggest to publish it with minor revisions.

R2Q1. Major concern: The authors have admitted that the PRECIS-2 tool does not appear to have been previously used to aid the design and conduct of, or rate, efficacy trials. It would be more convincible that author can either list some evidences or indicate some similar studies or close studies to demonstrate that the use of PRECIS-2 tools herein is proper.

R2A1. The reviewer raises a valid point. However, while the PRECIS-2 survey has not been used to evaluate efficacy trials to date, the PRECIS-2 survey and scoring system was designed to grade all trials in a continuum from very pragmatic (i.e., an effectiveness trial conducted under real-world conditions) to very explanatory (i.e., an efficacy trial conducted under ideal conditions). By definition, efficacy trials are very explanatory.  While the use of the PRECIS-2 survey in the literature has focused on the evaluation of pragmatic trials, there is no reason that a tool like the PRECIS-2 survey should not be used in efficacy trials too. This is what we believe is also novel about our paper. 

Minor revisions:

R2Q2. Some readers may not very familiar with PRECIS-2. It’s better to explain more in the introduction to help readers better understand.

R2A2. In the introduction we say: “We used the revised Pragmatic Explanatory Continuum Indicator Summary (PRECIS-2) to survey HAPIN investigators on the degree of pragmatism in nine domains of trial design and conduct (Loudon et al. 2015). PRECIS-2 is a tool to help trialists make design decisions consistent with the intended purpose of their trial. PRECIS-2 evaluates nine domains of trial design and conduct, namely eligibility, recruitment, setting, organization, flexibility in delivery, flexibility in adherence, follow-up, primary outcome and analysis, and uses a five-point Likert rank scale from very explanatory (1) to very pragmatic (5) to rank each of these domains. Therefore, efficacy trials would be expected to rank at or near 1 whereas effectiveness trials would be expected to rank at or near 5.”

Moreover, we provide a detailed description of the PRECIS survey in the Online supplement.

R2Q3. Line 136: How many control homes have LPG stove by themselves. That may important to know how their usual cooking practices are.

R2A3.  This is also an important question, and it may help to explain investigators’ perceptions about control group selection. These data were previously published in IJERPH. As per our previous analyses, none of the control homes in Rwanda or Guatemala had LPG stoves at baseline; about 1% and 6% of control homes in India and Peru, respectively, had LPG stoves at baseline (See Figure 6 in Quinn et al. IJERPH 2021. https://pubmed.ncbi.nlm.nih.gov/34886324/). We have added a sentence about this and a reference to this paper in the discussion, which reads as follows:

“Another condition that can affect measurement of a treatment effect is the choice of usual care for the control group (Thompson & Schoenfeld 2007). In HAP trials, usual care means allowing people to continue with their usual cooking practices. For example, 1% and 6% control homes in India and Peru used LPG stoves at baseline whereas none did in Rwanda and Guatemala (Quinn et al. 2021). The choice of usual care for the control group may have prompted some HAPIN investigators to consider the study setting less than very explanatory.”

R2Q4. Line 147: How is personal HAP and indirect assessment of HAP exposure in children measured?

R2A4. This is another important question, and it is explained in detail in Johnson et al. 2020 published in EHP (https://pubmed.ncbi.nlm.nih.gov/32347764/). We added this reference in the methods section; however, we do not think this information needs to be included in this paper since this paper is not intended to present HAP exposure results from the HAPIN trial

R2Q5. Line 184: How are the nine domains of design in PRECIS-2 chosen?

R2A5. The nine domains of design in PRECIS-2 were chosen after a careful analysis with expert panels. The creators of the PRECIS-2 survey ran a two-round Delphi communication with contact authors who had cited PRECIS but who had neither been involved in the original development of PRECIS nor evaluated the utility of PRECIS. They discussed how to improve PRECIS, focusing on issues raised by PRECIS users and brainstorming with trialists in Dundee. The Delphi results were the basis for a brainstorming meeting in Toronto involving some of the original developers of PRECIS and some of those who had undertaken methodological work using PRECIS, together with clinicians and policymakers. They user-tested candidate PRECIS-2 models, with 19 international trialists, on a one-to- one basis (in person or via Skype). PRECIS-2 was modified in response to user testing. This is outlined in detail in Loudon et al. BMJ 2015, which is cited in our manuscript.  We added a short description to the methods to explain how the domains in PRECIS-2 were selected, which reads as follows:

“The nine domains of design and conduct in PRECIS-2 were selected by an expert panel of trialists after careful analyses that included two rounds of the Delphi method with authors who were not involved in the original development of PRECIS, focus groups with trialists including the original developers of PRECIS, and an external validation with 19 international trialists (Loudon et al., 2015).”

R2Q6. Line 248: Authors compared ranking according to where investigators worked. Did author consider other factors such as age, sex, country, location? It would be better to show the interaction between these factors and comparison of rankings across these factors

R2A6. This is an excellent question, but we just did not have enough data to stratify by age, sex or setting (we attempted to do these stratifications, but the data was quite sparse).

Round 2

Reviewer 1 Report

The reviewer's observations were answered by the authors.